# Performance of Washing-Free Printing of Disperse Dye Inks: Influence of Water-Borne Polymers

**DOI:** 10.3390/polym14204277

**Published:** 2022-10-12

**Authors:** Ling Li, Runshan Chu, Qianxue Yang, Minhua Li, Tieling Xing, Guoqiang Chen

**Affiliations:** 1National Engineering Laboratory for Modern Silk, College of Textile and Clothing Engineering, Soochow University, 199 Renai Road, Suzhou 215123, China; 2National Innovation Center of Advanced Dyeing and Finishing Technology, Tai’an 271000, China

**Keywords:** inkjet printing, disperse dye, polyester fabric, water-borne polymer

## Abstract

Dye-containing wastewater discharge from the textile industry poses a serious pollution hazard that can be overcome by eliminating the washing step following the dyeing process. To study the washing-free printing of disperse dye ink, a number of water-borne polymers were selected and added to the ink, and the properties of the inks were discussed. By optimizing the ink formulation, printed fabrics with high color strength and color fastness were produced. The effects of the addition of polyvinylpyrrolidone (PVP), polyvinyl alcohol (PVA), and polyethylene glycol (PEG) on the ink jetting performance and printing performance were intensively investigated. The migration–diffusion–fixation behavior of disperse dyes in inks on the polyester fiber was explored. The disperse dye ink with 0.075 wt.% PVA exhibited the strongest migration–diffusion effect. The PVA ink exhibited excellent jetting performance and printing color fastness, and the printing color strength was better than that of the PVP and PEG ink. The addition of PVA increased the difference between the solubility parameter of the disperse dyes and ink system, which improved the migration of disperse dyes from the ink system to the polyester fabric. Meanwhile, PVA could form a protective layer on printed fabrics because of its excellent film-forming properties at room temperature. The washing-free inkjet printing method developed in this study provides a theoretical basis for screening water-borne polymers and an environmentally friendly pathway for the printing of textiles.

## 1. Introduction

Inkjet technology has opened up a new avenue for textile printing [1,2]. It has become one of the most promising technologies in textile printing due to its high printing accuracy, and its ability to meet market demand for small batches, multiple varieties, and fast delivery [3,4]. However, direct inkjet printing using disperse dye typically involves pretreatment of the surface of polyester fabric through physical or chemical methods [5], to ensure that there is a roughness or microstructure on the surface of the fabric, which reduces the diffusion and infiltration of the ink on the surface of the fabric, thereby improving the clarity of the printed pattern. The chemical modification method uses cationic treatment agents and water-borne polymers to pretreat the surface of polyester fabrics [6,7], and the physical method typically uses plasma technology that allows low consumption of water and production of no wastewater [8,9,10]. However, the pretreatment of polyester fabrics not only prolongs the printing process, but also consumes energy. In addition, to prevent the inkjet printing process from affecting the feel of the fabric, multiple washes are performed to remove floating color, resulting in a large volume of wastewater discharge and environmental pollution. In recent years, water shortages and pollution to waterways caused by industrial sewage discharge have become increasingly problematic, especially wastewater pollution from printing and dyeing enterprises [11,12]. Therefore, to develop a disperse dye that prints on polyester without pretreatment and requires only a short washing process, it is necessary to first investigate the ink formulation.

Digital inkjet printing ink has more stringent requirements than traditional printing paste in terms of particle size and suspension stability [13]. In addition to disperse dye as colorant, disperse dye ink contains various additives, which can be selected to develop an ink with the desired properties. The main components of ink are as follows: disperse dye (0.3–15 wt.%), dispersant, moisturizing agent (20–30 wt.%), surfactant (1–2 wt.%), defoamer (0.25 wt.%), pH adjuster (0.05 wt.%), viscosity adjuster, and deionized water (30–80 wt.%). Following the addition of ink components, interactions between different components affect the physical and chemical properties and jetting performance of the ink. The link between ink components and inkjet printing properties has been the subject of numerous studies. The results indicate that designing printing ink with consideration of the solubility parameters of reagents and HLB values of surfactants, and regulating the viscosity and surface tension of ink, can result in excellent printing quality [14]. In addition, adding water-borne polymers such as polyvinylpyrrolidone (PVP) to the ink can reduce the aggregation of dye molecules and result in an ink with more stable ejection and a higher application performance [15]. To date, research on disperse dye inks has mostly focused on the ink formulation and the interaction between components. Therefore, details of the migration mechanism of disperse dyes between ink and polyester remain elusive.

Owing to their suitability for a variety of applications, polyvinyl pyrrolidone (PVP), polyvinyl alcohol (PVA), and polyethylene glycol (PEG) are three of the most widely used polymer compounds. PVA, PVP, and PEG are non-toxic, biocompatible, and water-soluble synthetic polymers that are used in a range of technical applications [16]. A polymer surfactant, PVP can be used as a dispersant, emulsifier, thickener, or particle size regulator in different dispersion systems. PVA is one of several hydrogels used in biomedical applications with excellent film-forming, emulsifying, and adhesive properties. Through cross-linking, PVA can be converted to a three-dimensional polymer network, which swells to absorb a large volume of water [17]. PEG is formed via the polymerization of ethylene oxide with good biocompatibility and is one of the few synthetic polymers approved by the FDA [18,19]. Modification of nanoparticles using PEG can improve their stability and dispersibility, which prevents agglomeration [20].

In this study, the solubility parameters were used to evaluate the effect of water-borne polymers on dye migration in ink. Water-borne polymers (PVP, PVA, and PEG) were added to disperse dye inks and the optimal ink formulation was determined. Then, the ink was applied to polyester, and the influence of additives on washing-free ink printing was investigated by the solubility parameters of additives and the jetting and printing capabilities of the ink. This work provides a theoretical basis for screening water-borne polymers.

## 2. Experimental

### 2.1. Materials

C.I. Disperse Blue 79 (press cake) and C.I. Disperse Yellow 48 (press cake) were obtained from Annoqi Group Co., Ltd. (Shanghai, China). Ethylene glycol (EG) and PEG1000 were provided by Chinasun Specialty Products Co., Ltd. (Jiangsu, China). Fatty alcohol polyoxyethylene ether (Laureth-4) was supplied by Yuanye Biological Co., Ltd. (Shanghai, China). KYC-710 foam-free powder was purchased from Keying Chem Co., Ltd. (Hangzhou, China). PVP (K30), with a molecular weight of 58,000 g/mol, was provided by Dibai Biotechnology Co., Ltd. (Shanghai, China). PVA was obtained from Aladdin Biochemical Technology Co., Ltd. (Shanghai, China). Methylene dimethylnaphthalene sodium sulfonate (dispersant MF, industrial grade) was purchased from Yousuo Chemical Technology Co., Ltd. (Shandong, China). Triethanolamine (TEOA) was supplied by Shanghai Macklin Biochemical Co., Ltd. (Shanghai, China). Deionized water was prepared in our laboratory. Polyester fabrics (65 g/m^2^) were supplied by Hongda weaving factory.

### 2.2. Preparation of Disperse Dye Paste

Disperse dye (10 wt.%), dispersant MF (10 wt.%), KYC-710 (0.25 wt.%), and deionized water (79.75 wt.%) were weighed and stirred using a magnetic stirrer (Jiangsu Kexi Instrument Co. Ltd., Changzhou, China). After mixing, 100 g zirconia beads (0.4–0.6 mm) were added and the mixture was ground for 24 h with an omnidirectional planetary ball mill (Nanjing Chi Shun Technology Development Co. Ltd., Nanjing, China) at a speed of 600 rpm. The zirconia beads were filtered out with qualitative filter paper to obtain a dye paste.

### 2.3. Preparation of Disperse Dye Ink

As shown in Table 1, the prepared disperse dye paste was mixed with EG, defoaming agent, Laureth-4, water-borne polymers (PVP, PVA, and PEG) and deionized water. After being stirred until homogeneous, the ink was filtered through 0.2 μm filters. The product was maintained at pH 7–8 using TEOA. The inks using disperse blue dye paste were named PVP ink, PVA ink, PEG ink, and Control sample, respectively.

### 2.4. Printing

Monochrome block patterns and 353 μm-wide straight lines were printed on the polyester fabric and polyethylene glycol terephthalate (PET) film using an Epson R330 inkjet printer (Seiko Epson Corporation, Nagano, Japan). After printing, the fabric and PET film were dried at 90 °C for 1 min and thermofixation at 190 °C for 1 min.

### 2.5. Characterization of Disperse Dye Ink

The mean particle size of samples was measured by dynamic light scattering at 25 °C with a Malvern Zetasizer (Nano-ZS90, Malvern Instruments Co., Worcestershire, UK). The disperse dye ink was diluted 1000 times in deionized water before measurements were taken, and its zeta potential was measured using a Malvern Zetasizer at 25 °C. An automatic surface tension meter (DCAT21, KINO Scientific Instrument Inc., Charlotte, USA) was used to test the surface tension of the disperse inks. The measurements were repeated three times at 20 °C. The pH values of the disperse inks were obtained using a pH meter (PHS-3C, INESA Scientific Instrument Co. Ltd., Shanghai, China). The conductivity of the ink was detected by a conductivity meter (ST3100C, OHAUS Instrument Co. Ltd., Shanghai, China). The viscosity and rheological curve measurements were obtained using a rotational viscometer (RheolabQC, Anton Paar Instruments Co. Ltd., Shanghai, China) at 20 °C with a rotation speed of 60 rpm. The shear rate range for testing rheological curves is 10–1000 s^−1^.

TGA of water-borne polymers (PVP, PVA, and PEG1000) was carried out between 20 and 700 °C under a nitrogen atmosphere using a Q600 simultaneous thermal analyzer. Samples (5–10 mg) were analyzed in an alumina pan with a 20 °C/min heating rate.

### 2.6. Migration-Diffusion of Disperse Dyes

The printed PET film was cut into small pieces and then adhered to the glass slide with the cross-section visible. A three-dimensional super depth digital microscope (TD-SDDM, VHX-1000, Keyence Corporation, Osaka, Japan) was used to observe the spread of the disperse dye on the PET film.

### 2.7. Printed Fabric Characterization

The printed lines on the fabric were observed by TD-SDDM. The print quality of the output was assessed in terms of line width, edge blurriness, and edge raggedness. The rate of change (W) of the line width was calculated using Equation (1) (print linewidth, D_0_ = 353 μm; measured line width, D/μm; linewidth relative change, W/%). Larger W values indicate more significant ink permeation and worse pattern quality.
(1)$$W=\frac{\left(D-D_{0}\right)}{D_{0}}\times 100\%$$

The color strength (K/S) of the printed fabric was determined by an Ultra Scan Hunter Lab K/S spectrophotometer (USA) with a D65 light source and a spectral scan range of 350–700 nm. The fastness to washing and rubbing was evaluated according to the appropriate international standards GB/T3921-2008 (soaping), GB/T3920-2008 (rubbing), and relative references.

The quasi-static mechanical properties of a single yarn of pristine polyester fabric and fabric printed with PVP ink, PVA ink, PEG ink and control sample all measured 3 times by the Instron 3365 universal tester (Instron, Norwood, MA, USA) with a tensile speed of 0.04 mm/s and a gauge of 40 mm [21].

The bending length of the pristine and printed fabrics were tested by the YG(B)022D automatic fabric stiffness tester according to GB/T 18318.1-2009. For each sample, six specimens were tested, and average results were given.

## 3. Results and Discussion

### 3.1. Effect of Water-Borne Polymers on Washing-Free Printing Performance of Disperse Dye Inks

To achieve disperse dye washing-free printing technology, the diffusion of the dye from the printing ink into the fabric should be promoted as much as possible. The solubility parameter (δ) plays a crucial role in the migration–diffusion of disperse dyes. When the solubility parameters of ink additives are similar to that of the dye, the components are highly compatible, which leads to inks with high stability and longevity. Unfortunately, such ink lacks the ability to promote the dyeing of fabrics. To improve the migration capabilities of dyes in ink, large differences between the solubility parameters of the disperse dye and that of the water-borne polymers are preferable [22].

Table 2 lists the solubility parameters of the disperse dyes and additives [23,24,25]. Based on these data, we carefully considered the influence of these water-borne polymers on the ink properties and printing performance and selected additives that would endow the ink with desirable characteristics.

### 3.2. Physicochemical Properties of Disperse Inks

To ensure that the ink demonstrates good application performance, it must have good physicochemical properties. As the aperture of the piezoelectric printer head is approximately 20–50 μm [26], the particle size of the disperse dye ink should be below 200 nm to avoid clogging of the printer head [27]. The particle size of disperse ink is approximately 180–200 nm. Another factor that can cause nozzle clogging is the salt content of the ink, which is characterized by the electrical conductivity. If the salt content of the ink is too high, crystallization may occur and affect the ink jetting. Generally, the conductivity of the printer ink is required to be below 104 μS/cm [28]. To avoid corrosion of the printer head, the pH value of printing ink was adjusted and maintained in the range 7–9 by pH regulator (TEOA) [27]. The zeta potential is the potential of the particles at the shear plane, which characterizes the dispersion stability of the liquid. The greater the absolute value of the zeta potential, the greater is the repulsive force between the colloidal particles, and the more stable the colloidal system. Typically, when the absolute value of the zeta potential is between 40 and 60 mV, the colloid has high stability, and the particle collision and agglomeration are significantly reduced. The formation of ink droplets during inkjet printing is related to the surface tension and viscosity. Improper performance leads to the formation of “satellite ink droplets,” which affect the definition of the printed pattern outline [29,30]. In addition, when the surface tension and viscosity are too low, pattern penetration occurs [31]. Commercial inkjet inks for textile printing typically have surface tension values between 21 and 48 mN/m. Table 3 shows that the surface tensions of the four inks were adjusted by surfactant (Laureth-4) to the normal range of commercial inks. Physicochemical properties of disperse yellow dye inks are displayed in Appendix A.

### 3.3. Rheological Properties of the Disperse Ink

Ink rheology is an important index to evaluate ink performance. It depends on the solid content, solvent ratio, and additive. Solvents contribute significantly to the rheological properties of inks. The viscosity and viscoelasticity of inks alter in response to the addition of various polymers, which can change the rheology and affect the jetting performance of the ink.

Figure 1a shows the viscosity curves of ink solutions at different shear rates. The viscosities of the inks after adding 0.5% PVP, 0.075% PVA, and 2% PEG were 2.31, 2.31, and 2.56 mPa∙s, respectively. The reason is that the PVA and PEG have intramolecular/intermolecular hydrogen bond interactions due to their hydroxyl groups, and the microstructure of PVP exhibits a honeycomb structure. Both hydrogen bonding and honeycomb structure can restrict water in the matrix, reduce the fluidity of the ink, and improve the viscosity of the system. The ink additive with 2% PEG 1000 has a high viscosity. Appendix A shows the correlation between shear rate and viscosity of disperse blue dye inks at different temperatures, and the viscosity are summarized in Appendix A.

Figure 1b shows the shear stress of the ink as a function of shear rate. Regardless of the type of ink used, the shear stress increased linearly with the shear rate. Therefore, the inks in the experiment are all typical Newtonian fluids [26]. The correlation between shear rate and viscosity and the correlation between shear rate and shear stress of disperse yellow dye inks with different additives are displayed in Appendix A.

### 3.4. Thermal Analysis

In Figure 2, the TGA thermograms of three water-soluble polymers are compared. The polymers lose weight during TGA in two or three main steps. The first step is below 250 °C, and can be ascribed to the loss of moisture, and the other steps are above 250 °C, and can be associated with the degradation of the polymers. The decomposition temperatures of PVP, PVA, and PEG are 337 °C, 263 °C, and 288 °C, respectively. This demonstrates that the polymers have high stability and can withstand high temperatures during thermofixation.

### 3.5. Migration and Diffusion of Disperse Dyes in PET Film

Polyester is a thermoplastic fiber with high crystallinity. The diffusion of disperse dyes into PET fibers belongs to the free volume model. When the fiber is heated above the glass transition temperature, the movement of macromolecular segments of the fiber is intensified, the intermolecular gap is enlarged, and dye can diffuse into the fiber in the form of single molecule [32].

The disperse dye inks with different additives were jetted onto the surface of the PET films, and the migration and diffusion of disperse dyes in the film were observed after thermofixation. As shown in Figure 3, the disperse dyes exhibited different degrees of diffusion on the PET films during thermofixation. Relative to that of the control sample, the migration of the ink with added water-borne polymer in the PET film was enhanced, and the effect was the most obvious for the PVA ink. This may be because PVA and ink have the biggest difference in solubility parameters, and PVA ink system has the worst compatibility. More dyes in the ink were released from the ink system and migrate into the fibers during thermofixation. The cross-section images of PET films printed with disperse yellow dye inks are displayed in Appendix A.

### 3.6. Surface Morphology of the Printed Polyester Fiber

Figure 4 shows an SEM images of pristine polyester fabric and polyester fabric printed with PVP ink, PVA ink, PEG ink and control sample. There are films on the surface of the printed fabrics, which help to fix the disperse dyes on the fabric and improve the color fastness. In addition, the surface of the fabric printed with PVA ink is more consistent and smoother. This may be attributable to the good film-forming properties of PVA at room temperature. And the SEM images of fabrics printed with disperse yellow dye inks are displayed in Appendix A.

### 3.7. Injection Performance of Disperse Ink

When the ink in the ink chamber is subjected to a certain pressure, a small droplet is formed at the nozzle due to the action of pressure and surface tension, resulting in the ejection of an ink droplet [33,34,35]. In addition to the size and waveform of the driving voltage, the properties of the ink itself, such as viscosity, surface tension, and rheological properties, significantly influence the speed and continuity of droplet ejection [26,36]. The formation of stable single droplets during jetting is essential to produce high-quality inkjet-printed products. The speed of the droplets, the quality of the droplet morphology, and the presence or absence of satellite droplets affect the quality of the final inkjet print. For optimal jetting performance, the viscosity of ink should be kept low. Additionally, the surface tension of the ink needs to be low enough for it to moisten the capillary channel and flow through the nozzle.

Figure 5 shows a photo of the ink droplet morphology at intervals of 2 μs during ejection from the nozzle. Droplet formation involves jetting, stretching, necking, breaking, the creation of satellite dots, and the single droplet coincidence of the ink fluid. The formation of ink droplets occurs over several stages. Pulsing occurs and the droplets are squeezed out of the nozzle. The continuous action of pressure causes the formation of ink droplets with a rounded front end and an elongated rear end over a period of 22–32 μs. Subsequently, due to the change in pressure waveform from the ink chamber, surface tension causes the ink drops to break away from the print head in 32–36 μs and form a ligament. The broken ink droplets recoil and break to form satellite ink droplets. When the velocity of satellite ink droplet is higher than that of the main ink droplet, the two droplets gradually combine to form a stable, spherically shaped ink droplet.

Figure 5 shows that all three water-borne polymer additives can form stable ink droplets, and the broken satellite ink droplets approach the main ink droplets to form spherical droplets. Among the additives, PEG has the most significant impact on the ink, and the satellite ink droplets take the longest time to reach the main ink droplets, which may cause the formation of oblique jets and affect the printing accuracy.

### 3.8. Printing Performance of Disperse Ink

Table 4 shows the color strength and rubbing fastness data of the printed samples. The K/S values of the samples are 6.82 (PVP ink), 7.59 (PVA ink), 6.93 (PEG ink), and 6.95 (control sample). The dry rubbing fastness and wet rubbing fastness of inks with additives reached grade 4 and grade 3–4, respectively, demonstrating an improvement in relation to the control samples. The washing fastness of the samples reached grade 4–5. This is probably because the ink forms a film on the surface of the fabric after thermofixation to protect the printed fabric (Figure 5). And the color yield and color fastness of the printed polyester fabric using disperse yellow dye inks are displayed in Appendix A.

Meanwhile, printing clarity tests were carried out on 4 types of self-made inks. Table 5 shows the line width and linewidth relative change (W) of the printed samples.

The water-borne polymer-added samples demonstrated better print clarity than the control sample. This is because the addition of water-borne polymers facilitates the formation of hydrogen bonds between ink components and forms an interpenetrating network structure, which increases the viscosity of disperse dye inks and reduces the fluidity [37]. Among the inks, the PVA ink exhibited excellent printing clarity. It is clear that there was a gap between the printed line widths in the warp and weft directions on the fabric. This stems from the difference in wicking rates along the warp and weft directions [38]. And the Linewidth and linewidth relative change (W) of fabric printed with disperse yellow dye inks are listed in Appendix A.

For quasi-static tests, it can be seen from Table 6 that the breaking strength of the fabrics printed with different formulations was not much different from that of the pristine fabrics. It shows that printing had no significant effect on tensile breaking strength. However, the elongation at break of the fabrics printed with PVP ink, PVA ink, PEG ink, and control sample was 27.99%, 26.65%, 27.92%, 27.55%, respectively, which were improved to some extent compared with the pristine fabrics (25.21%). This may be due to the fact that the ink forms a film on the surface of the fabric, which can protect the outer layer of the yarn. The quasi-static stress-displacement curves of yarns of pristine polyester fabric and fabric printed with PVP ink, PVA ink, PEG ink, and control sample can be seen in Figure 6.

Table 7 shows the bending length of printed polyester fabric in warp and weft directions. After ink printing, the bending length of the fabric was increased. This should be attributed to the printed coating on the fabric (Figure 4), which prevented the mutual sliding of fibers and yarns, resulting in adverse effects on fabric softness.

## 4. Conclusions

For polyester fabric, a washing-free printing technique was created to help reduce chemical dosage and eliminate the need for water. Disperse dye inks with 0.5 wt.% PVP, 0.075 wt.% PVA, and 2 wt.% PEG added demonstrated good jetting performance and improved the printing color strength and color fastness. The dry and wet rubbing fastness of the water-borne polymer-added samples were grade 4 and grade 3–4, respectively, while the control sample is grade 3–4 and grade 3. With reference to the solubility parameter, PVA was selected as the best water-borne polymer additive because of its superior ink jetting and printing performance. In the process of ink droplet ejection, the ink droplets were ejected simultaneously at 22 μs, and the time for the satellite droplets of the PVA ink to reach the main ink droplet was shortened to 46 μs, while the time for the control sample was 52 μs. The PVA ink demonstrated better migration and diffusion ability and a higher color strength than those made with PEG and PVP. The K/S values of the PVA ink reached the highest (7.59) compared to other inks. The disperse dye ink formulation developed in this study is beneficial to preserve scarce freshwater resources. By reducing the pre-printing treatment and subsequent washing processes, the developed method significantly reduces the negative environmental impact of printing, opening up a new avenue for polyester short-run printing. Disperse dye direct jet washing-free printing for polyester is a promising technology for improving sustainable practices in the textile industry.

## Figures and Tables

**Figure 1 polymers-14-04277-f001:**
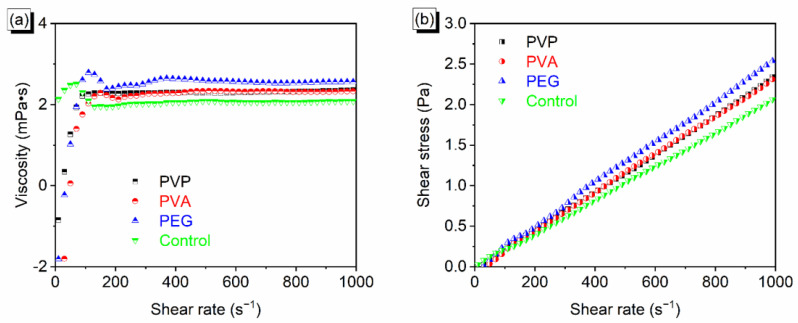
(**a**) Correlation between shear rate and viscosity of inks with different additives, and (**b**) correlation between shear rate and shear stress.

**Figure 2 polymers-14-04277-f002:**
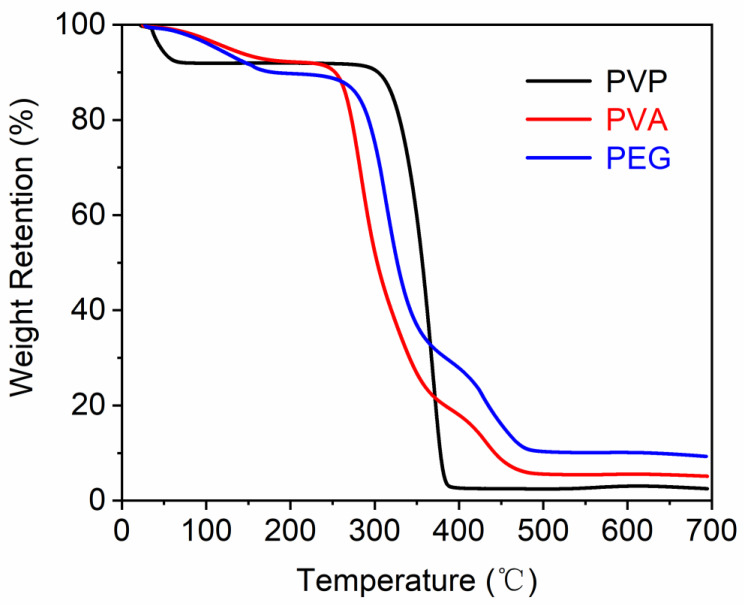
TG curves of PVP, PVA, and PEG.

**Figure 3 polymers-14-04277-f003:**
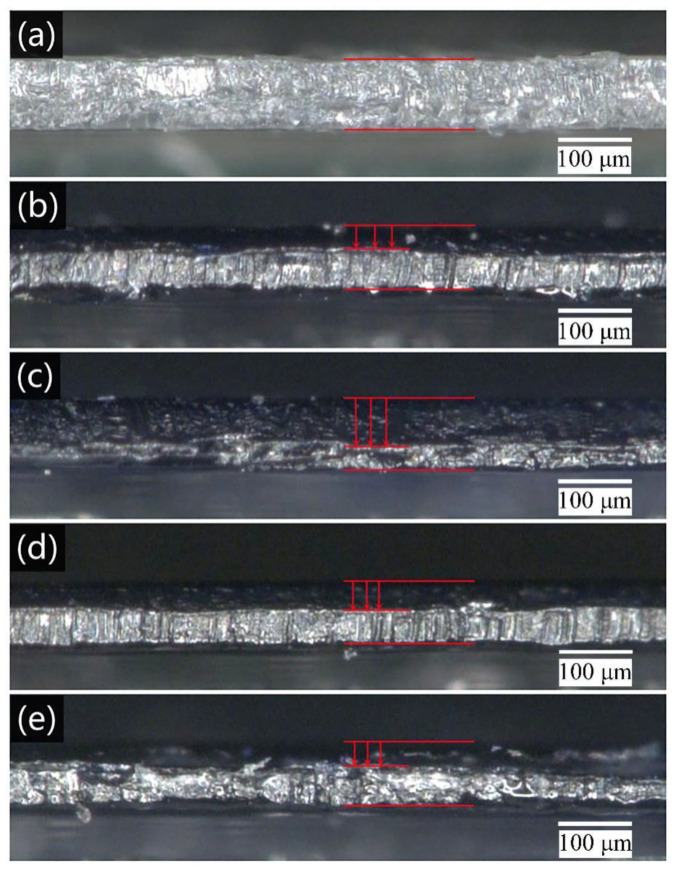
Cross-section images of printed PET films: (**a**) pure PET film, (**b**) PET film printed with PVP ink, (**c**) PET film printed with PVA in, (**d**) PET film printed with PEG ink, and (**e**) PET film printed with control sample.

**Figure 4 polymers-14-04277-f004:**
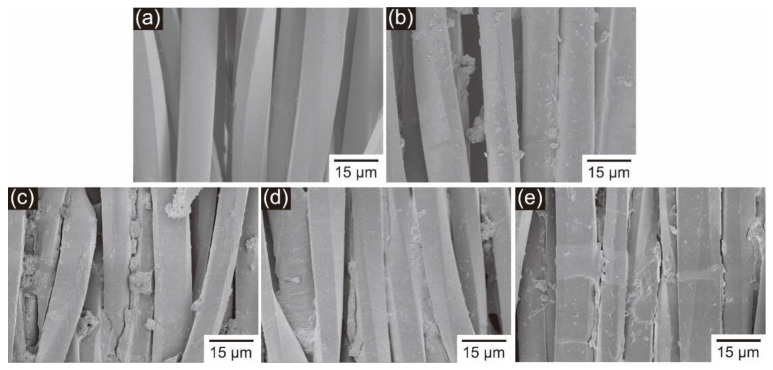
SEM images of fabrics: (**a**) polyester fabric, (**b**) polyester fabric printed with control sample, (**c**) polyester fabric printed with PVP ink, (**d**) polyester fabric printed with PVA ink, and (**e**) polyester fabric printed with PEG ink.

**Figure 5 polymers-14-04277-f005:**
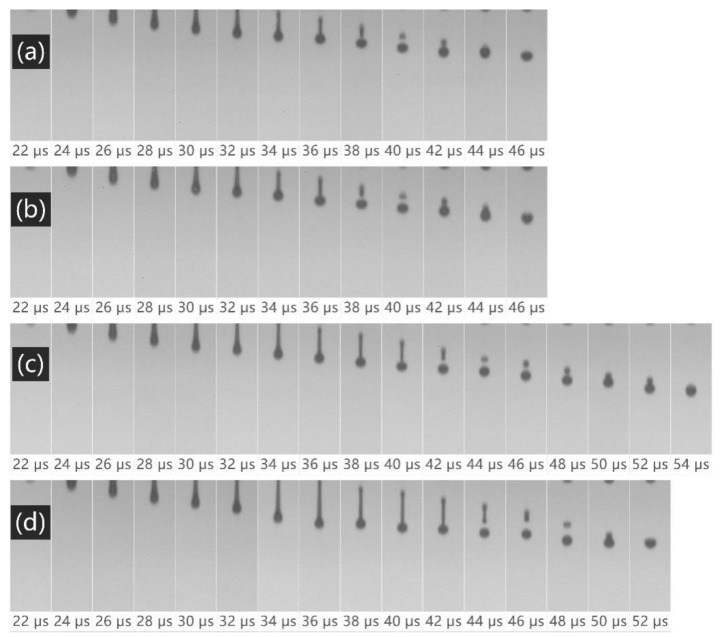
Ejection properties of disperse dye ink: (**a**) PVP ink, (**b**) PVA ink, (**c**) PEG in, (**d**) control sample.

**Figure 6 polymers-14-04277-f006:**
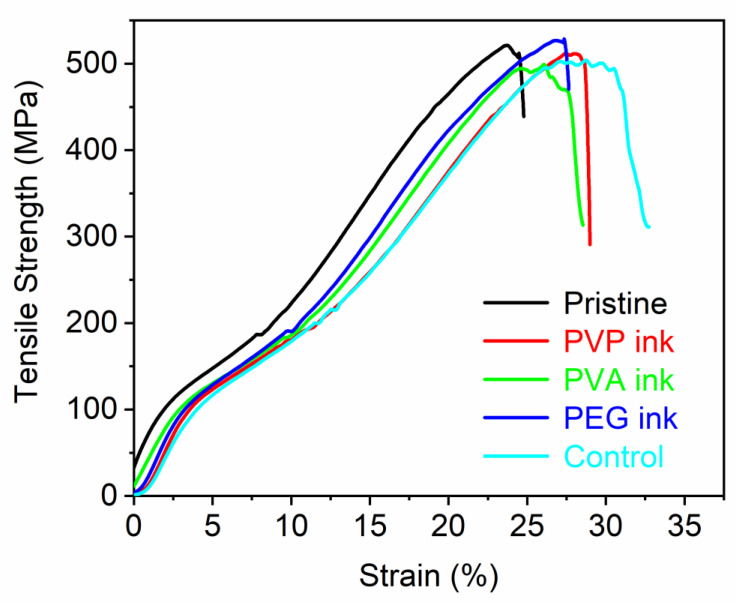
Quasi-static stress-displacement curves of yarns of pristine polyester fabric and fabric printed with PVP ink, PVA ink, PEG ink, and control sample.

**Table 1 polymers-14-04277-t001:** The formulation of disperse dye inks.

Sample	Disperse Dye Paste (wt.%)	EG (wt.%)	KYC-710 (wt.%)	Laureth-4 (wt.%)	PVP K30 (wt.%)	PVA5% (wt.%)	PEG1000 (wt.%)	TEOA (wt.%)	Deionized Water (wt.%)
PVP	30	20	0.25	0.5	0.5	/	/	0.05	48.7
PVA	30	20	0.25	0.5	/	0.075	/	0.05	49.125
PEG	30	20	0.25	0.5	/	/	2	0.05	47.2
Control	30	20	0.25	0.5	/	/	/	0.05	49.2

Abbreviations: EG, ethylene glycol; KYC-710, defoamer; Laureth-4, fatty alcohol polyoxyethylene ether; PEG, polyethylene glycol; PVA, polyvinyl alcohol; PVP, polyvinylpyrrolidone; TEOA, triethanolamine.

**Table 2 polymers-14-04277-t002:** Solubility parameters (δ) of the disperse dyes and additives.

Additives and Dye	δ (J/cm^3^)^0.5^
Disperse Dye	26.14
PVP	24.30
PVA	33.06
PEG	22.87
Water	44.91

**Table 3 polymers-14-04277-t003:** Physicochemical properties of disperse inks.

Sample	Particle Size (nm)	Zeta Potential (mV)	pH	Conductivity (μS/cm)	Surface Tension (mN/m)	Viscosity (mPa∙s)
PVP	184.9	−53.6	8.17	6.78	29.78	2.31
PVA	185.0	−54.8	8.23	6.91	30.83	2.31
PEG	188.1	−62.3	8.09	6.41	29.94	2.56
Control	188.4	−54.6	8.27	6.82	30.36	2.06

**Table 4 polymers-14-04277-t004:** Color yield and color fastness of the printed polyester fabric using disperse dye inks with different additives and control sample.

Sample	K/S	Rubbing Fastness		Washing Fastness
		Dry	Wet	
PVP	6.82	4	3–4	4–5
PVA	7.59	4	3–4	4–5
PEG	6.93	4	3–4	4–5
Control	6.95	3–4	3	4–5

**Table 5 polymers-14-04277-t005:** Linewidth and linewidth relative change (W) of printed fabric.

Sample	Warp			Weft		
	Printing Pattern	Linewidth (μm)	W (%)	Printing Pattern	Linewidth (μm)	W (%)
PVP	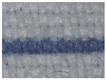	418.09	18.44	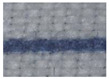	370.42	4.93
PVA	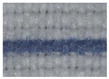	412.21	16.77	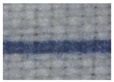	371.05	5.11
PEG	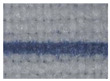	421.99	19.54	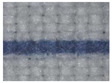	385.39	9.18
Control	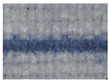	428.50	21.39	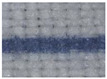	401.48	13.73

**Table 6 polymers-14-04277-t006:** Breaking strengths and breaking elongation of yarns of pristine polyester fabric and fabric printed with PVP ink, PVA ink, PEG ink, and control sample.

Sample	Breaking Strengths (cN)	Breaking Elongation (%)
Pristine	309.80 ± 1.52	25.21 ± 1.20
PVP ink	307.07 ± 2.52	27.99 ± 1.14
PVA ink	299.97 ± 2.87	26.65 ± 0.36
PEG ink	310.39 ± 2.32	27.92 ± 0.90
Control	301.50 ± 2.86	27.55 ± 1.05

**Table 7 polymers-14-04277-t007:** Bending length of the pristine and printed fabrics.

Sample	Bending Length (mm)	
	Warp	Weft
Pristine	31.9 ± 0.7	30.7 ± 4.7
PVP ink	40.3 ± 1.0	39.4 ± 1.9
PVA ink	36.6 ± 1.8	32.5 ± 1.5
PEG ink	38.0 ± 2.1	35.8 ± 0.7
Control	34.6 ± 2.7	32.3 ± 1.6

## Data Availability

The raw data presented in this study are available on request from the corresponding author.

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
