# Peer review of "Performance of Washing-Free Printing of Disperse Dye Inks: Influence of Water-Borne Polymers"

_polymers, 2022, doi:10.3390/polym14204277_

Round 1
Reviewer 1 Report
i recommend the authors to :
- add statical analysis
- carry a durability test
Reviewer 2 Report
Polymers 1912070
Washing-free performance of ..
By Li et al.
Authors describe as study to modify ink-jet printing inks for polyester printing with addition of polymer materials e.g. PVP, PVA, and PEG.
Besides physico chemical characterisation of the inks and the disperse dyes also the printing performance has been studied and the colour strength and fastness of the printings have been investigated.
The study represents an extensive investigation of the influence of polymer addition to the performance of an ink-jet formulations, however a number of points require careful consideration.
Authors have to explain the novelty of their findings in the light of their own (not cited) publication which is very near to the content of the present manuscript:
Preparation of disperse inks for direct inkjet printing of non-pretreated polyester fabrics, DOI: 10.1039/C9RA01999E (Paper) RSC Adv., 2019, 9, 19791-19799
Authors state that their printing does not require washing, did they test the fastness on the printed lines or did they print a full square in the size of the fastness samples?
How the polyester fabric was pretreated (pre-washed?) or do they print on the spin-finish of the fibre spinning?
Line 194 when the polymers interact with the solution, why viscosity does not change much?, how then the effect of the polymer is explained
Line 213: PET flakes? Unclear
Line 214. Here films were printed (no description in the experimental)
Line 216: baking? Does this mean thermofixation? Pls use correct terminology
Line 217 – 221: unclear, what means migration ability, reduced affinity between ink system?
Table 3: viscosity of PEG sample 2.56, while in line 194 2.43 mPa s
Line 194-195 hydrogen bonding between molecular chains of the solution? Unclear
Figure 3: scale bar is missing
Figure 4: reference print control is missing
Figure 5 which picture shows which formulation
Line 277 – 279 unclear, what means reduced fluidity, is this the viscosity?
Line 65: are PVA, PVP and PEG biodegradable?
Line 208: pls check significant digits of decomposition temperatures
Conclusions line 292 – 297 these arguments are speculative and not supported by the data given. Here authors should use calculations and comparison to the state of the art technology to support the statements.
Reviewer 3 Report
The manuscript is well desinged and has utilized various appropriate characterization techniques. The manuscript is overall well written. I just have couple points:
1. Why authors chose particular formulation for disperse dye paste (section 2.2), please provide reference
2. What was the refractive index value used for disperse dye index in DLS measurement? What is particle size distribution curve look like?
